# Controllable and Lossless
# Non-Autoregressive End-to-End Text-to-Speech

## Abstract

Some recent studies have demonstrated the feasibility of single-stage neural text-to-speech, which does not need to generate mel-spectrograms but generates the raw waveforms directly from the text. Single-stage text-to-speech often faces two problems: a) the one-to-many mapping problem due to multiple speech variations and b) insufficiency of high frequency reconstruction due to the lack of supervision of ground-truth acoustic features during training. To solve the a) problem and generate more expressive speech, we propose a novel phoneme-level prosody modeling method based on a variational autoencoder with normalizing flows to model underlying prosodic information in speech. We also use the prosody predictor to support end-to-end expressive speech synthesis. Furthermore, we propose the dual parallel autoencoder to introduce supervision of the ground-truth acoustic features during training to solve the b) problem enabling our model to generate high-quality speech. We compare the synthesis quality with state-of-the-art text-to-speech systems on an internal expressive English dataset. Both qualitative and quantitative evaluations demonstrate the superiority and robustness of our method for lossless speech generation while also showing a strong capability in prosody modeling.

## 1  Introduction

With the rapid development of deep learning, neural text-to-speech (TTS) systems can generate natural and high-quality speech and thus have drawn much attention in the machine learning and speech community. TTS is a task that aims at synthesizing raw speech waveforms from the given source text. Most previous neural TTS systems' pipelines are two-stage. The first stage is to generate intermediate speech representations (e.g., mel-spectrograms) autoregressively [36, 29, 24, 19] or non-autoregressively [27, 26, 14] from input text. The second stage is to synthesize speech waveforms from the generated intermediate speech representations using a vocoder [10, 22, 25, 38, 33]. These systems with two-stage pipelines can synthesize high-quality speech but still have drawbacks because they need sequential training or fine-tuning [15]. In addition, the use of predefined intermediate representations prevents further improvement in overall performance, as two system components can not be jointly trained and connected by learned intermediate representations.

Recently, several works (FastSpeech 2s [26], EATS [7], VITS [15]) have proposed parallel end-to-end TTS models that generate raw waveforms directly from input text in a single stage. FastSpeech 2s introduces explicit pitch and energy as mel-spectrogram decoder conditions to alleviate the one-to-many mapping problem in the TTS system. However, it needs to extract these handcrafted features in

advance, complicating the training pipeline. Moreover, FastSpeech 2s only models pitch and energy but does not disentangle other prosody features from the speech. EATS and VITS can synthesize high-quality speech, but they do not disentangle prosody information from speech, so they can not achieve prosody modeling and control.

To solve the problem that previous single-stage parallel end-to-end TTS models do not model the general prosody, we propose the **C**ontrollable and **LO**ssless **N**on-autoregressive **E**nd-to-end TTS (CLONE), which contains some carefully designed components to disentangle and model the general prosody from speech.

Firstly, to better solve the one-to-many mapping problem, we need to model the information variance other than text in speech. We propose an implicit phoneme-level prosody latent variable modeling instead of only explicitly modeling pitch and energy in FastSpeech 2s. The phoneme-level prosody latent variable models general prosody in the speech in a unified way without supervision. Specifically, we assume that prosody follows a normal distribution and use a variational autoencoder [16] (VAE) with normalizing flows [28, 6] to model it, which enhances the modeling ability of pure VAE and enables better modeling of prosody information that has extremely high variance. We propose a prosody predictor to predict the prior distribution of phoneme-level prosody latent variable from the input phoneme, which enables end-to-end synthesis as a TTS system.

In addition, we carefully study the problem of unsatisfactory high-frequency information generation in single-stage end-to-end speech synthesis, which is caused by the lack of the supervision of ground-truth acoustic features during training. To enhance the learned intermediate representation, we propose the dual parallel autoencoder (DPA) that consists of two parallel encoders (the acoustic encoder and the posterior wave encoder) and a wave decoder. DPA uses ground-truth linear spectrograms to regularize the learned intermediate representations for efficient learning. Besides, we introduce the multi-band discriminator (MBD) that significantly speeds up model convergence and improves generation quality. DPA and MBD enable CLONE to synthesize high-quality speech at the lossless high sampling rate (48 kHz) with better high-frequency information.

We conduct experiments on our private speech datasets. The results of extensive evaluations show that CLONE outperforms SOTA two-stage and single-stage TTS models [29, 26, 15] in terms of speech quality. In addition, CLONE can synthesize lossless speech at 48 kHz with better speech quality. Furthermore, we demonstrate that CLONE can model and control prosody by the phoneme-level prosody latent variable and generate speech with appropriate prosodic inflections. We attach audio samples generated by CLONE at `https://cloneurips2022.github.io/CLONE/`.

## 2 Related Work

**Text-to-Speech**   Text-to-Speech (TTS), which aims to synthesize intelligible and natural speech waveforms from the given text, has attracted much attention in recent years. Specifically, the neural network-based TTS models [36, 29, 27, 26, 22] have achieved tremendous progress. The quality of the synthesized speech is improved a lot and is close to that of the human counterpart. The previous prevalent methods are two-stage. The general pipeline of two-stage methods is: first, generate the acoustic features (e.g., mel-spectrograms) from text autoregressively [36, 29, 24, 19, 34] or non-autoregressively [27, 26, 14, 23], then synthesize the raw waveforms conditioned on the acoustic features [10, 22, 25, 18]. Recently, several single-stage end-to-end TTS models [26, 7, 15] have been proposed to generate raw waveform directly from the text. Among them, VITS [15] outperforms the two-stage models due to the advantages of learned intermediate speech representations obtained by fully end-to-end training. However, these single-stage methods have poor controllability over the prosody of synthesized speech. Specifically, EATS [7] and VITS cannot control prosody. FastSpeech 2s [26] can only control some pre-defined prosodic features (i.e., pitch and energy), and these features are required to be extracted in advance. Unlike the aforementioned single-stage models, by utilizing a conditional VAE with normalizing flows, CLONE achieves high controllability over the general phoneme-level prosody of synthesized speech.

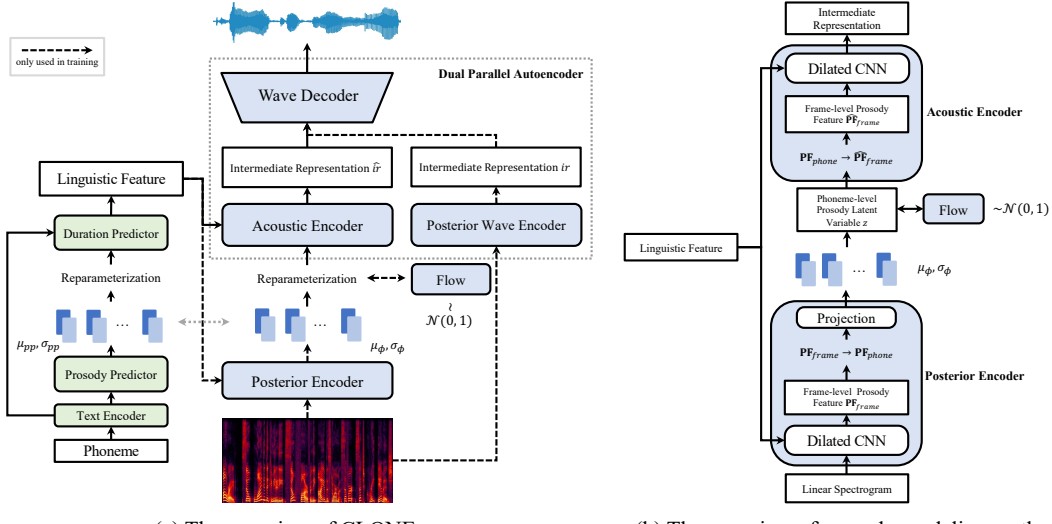

(a) The overview of CLONE

(b) The overview of prosody modeling method

Figure 1: The overview of CLONE and prosody modeling method.

**Prosody Modeling**  Many previous works have focused on learning underlying non-textual information (e.g., style and prosody) in speech. In particular, some works [30, 37] introduce a reference embedding to model style and prosody. [30] extracts a prosody embedding from a reference spectrogram, and [37] models a reference embedding as a weighted combination of a bank of learned embeddings. Some works [1, 39] use the variational autoencoder (VAE) to model latent representations for styles and prosody of speech. Specifically, [32] uses multi-level VAE to model fine-grained prosody at the phoneme and word level in an autoregressive way. [39] and [12] integrate VAE with Tacotron 2 for better style modeling. Some works [8, 11] use GMM based mixture density network to model prosodic information at phoneme and word levels. Unlike the previous models, CLONE adopts a conditional VAE with normalizing flows for the phoneme-level prosody modeling to a single-stage parallel TTS system. Inspired by [28, 5, 40] that improve the expressive capability of prior and posterior distribution with normalizing flows, we add normalizing flows to enhance the representation power of our prior distribution for better prosody modeling. Furthermore, the prosody predictor enables CLONE to predict the prior distribution of prosody directly from the text and control the prosody of generated speech during inference without the need for manually adjusting the sampling points [39] or a reference speech [37] or other modality input (e.g., video [13]).

## 3  CLONE

### 3.1  Overview

The overall model structure of CLONE shown in Figure 1a can be regarded as a conditional VAE. Firstly, the posterior encoder converts the input spectrograms to a sequence of phoneme-level prosody latent variable $z$. Then, the acoustic encoder transforms the prosody latent variable $z$ into the learnable intermediate representations conditioning on linguistic features. Finally, the wave decoder predicts the waveforms from the learnable intermediate representations. The objective of CLONE is to maximize the evidence lower bound (ELBO) of the intractable marginal log-likelihood of data $\log_\theta(x \mid c)$:

$$\text{ELBO} = \mathbb{E}_{q_\phi(z|x,c)} \left[\log p_\theta(x \mid z, c)\right] - D_{kl}\left(q_\phi(z \mid x, c)\|p_\theta(z \mid c)\right), \tag{1}$$

where $c$ and $z$ denote the linguistic feature and the phoneme-level prosody latent variable respectively, $q_\phi(z \mid x, c)$ is the approximate posterior distribution of $z$ given a data point $x$ and condition $c$, $p_\theta(x \mid z, c)$ is the likelihood function of $x$ given $z$ and $c$, and $p_\theta(z \mid c)$ is the prior distribution of $z$ given $c$.

The training loss of CLONE is the negative ELBO, which consists of the reconstruction loss ($-\mathbb{E}_{q_\phi(z|x,c)}\left[\log p_\theta(x \mid z, c)\right]$) and the KL divergence ($D_{kl}(q_\phi(z \mid x, c)\|p_\theta(z \mid c))$). The details of the reconstruction loss and the KL divergence are described in Section 3.4.2 and Section 3.2, respectively.

## 3.2 Prosody Modeling

To model the prosody of speech more appropriately, we determine to model the phoneme-level prosody rather than the frame-level prosody or the word-level prosody. Because the frame-level prosody causes severe linguistic information leakages during training, and the granularity of the word-level prosody is too large to reflect the details of the prosody well. Since the input linear spectrogram is at the frame level, we need to convert the frame-level prosody feature to the phoneme level. We use the obtained duration of phonemes to construct the hard alignment matrix representing the correspondence between phonemes and spectrogram frames and convert the frame-level prosody feature to the phoneme-level prosody feature by the matrix as follows:

$$\mathbf{PF}_{phone} = \mathrm{diag}(\mathbf{s}) \cdot \mathbf{A} \cdot \mathbf{PF}_{frame}, \tag{2}$$

where $\mathbf{PF}_{phone} \in \mathbb{R}^{T_p \times d}$ and $\mathbf{PF}_{frame} \in \mathbb{R}^{T_f \times d}$ denote the phoneme-level prosody feature and the frame-level prosody feature, respectively, $\mathbf{A} \in \mathbb{R}^{T_p \times T_f}$ denotes the hard alignment matrix, and $\mathbf{s} \in \mathbb{R}^{T_p}$ denotes the inverse of the duration of phonemes ($s_i = 1/\sum_{j=1}^{T_f} a_{ij}$).

The mean and standard deviation of the approximate posterior distribution $q_\phi(z \mid x, c)$ are predicted from the obtained phoneme-level prosody feature and linguistic feature. We obtain the linguistic feature $c$ by expanding the output of the text encoder according to the phoneme duration. The length of the linguistic feature is the same as the number of frames in the spectrogram.

The formula for KL divergence is as follows:

$$\mathcal{L}_{kl} = D_{kl}\left(q_\phi(z \mid x, c)\|p_\theta(z \mid c)\right) = \mathbb{E}_{q_\phi(z|x,c)}\left[\log q_\phi(z \mid x, c) - \log p_\theta(z \mid c)\right]. \tag{3}$$

Unlike traditional VAE, we assume that the approximate posterior distribution of the phoneme-level prosody latent variable $z$ is a normal distribution rather than a standard normal distribution, i.e., $q_\phi\left(z \mid x, c\right) = \mathcal{N}\left(z; \mu_\phi\left(x, c\right), \sigma_\phi\left(x, c\right)\right)$. As a TTS model, we want to control the phoneme-level prosody explicitly. If $z$ follows the standard normal distribution, the prosody variation is implicitly determined in the random sampling process, which is not desired. Thus, a normal distribution with variable mean and variance is a better choice. It enables the prosody variation to be contained in the mean and variance of normal distribution so that the corresponding prosody can be determined by predicting the mean and variance. In addition, compared with standard normal distribution, normal distribution is more complex, which increases the prosody modeling ability to obtain diverse prosody variation.

We also need to increase the expressiveness of the prior distribution to match the posterior distribution. Therefore, we apply normalizing flows, which enable an invertible transformation from a simple standard normal distribution into a more complex prior distribution following the rule of change-of-variables:

$$p_\theta(z \mid c) = \mathcal{N}\left(f_\theta(z); \mathbf{0}, \boldsymbol{I}\right)\left|\det \frac{\partial f_\theta(z)}{\partial z}\right|, \tag{4}$$

where $f_\theta$ denotes the normalizing flow. After the reparameterization of VAE, we get the phoneme-level prosody latent variable $z$ which represents the phoneme-level prosody of the speech. We need to convert phoneme-level $z$ to frame-level variable $\widehat{\mathbf{PF}}_{frame}$ to match the length of the linguistic feature as follows:

$$\widehat{\mathbf{PF}}_{frame} = \mathbf{A}^\top \cdot z, \tag{5}$$

where $\mathbf{A}^\top$ is the transposed matrix of $\mathbf{A}$.

## 3.3 Prosody Predictor

We propose a prosody predictor to model the correspondence between phoneme and phoneme-level prosody. The prosody predictor can predict the mean and variance of $q_\phi\left(z \mid x, c\right)$ from the output

of the text encoder. Therefore, CLONE can predict phoneme-level prosody from text input in the inference stage. During inference, there are three modes to generate highly natural speech with suitable prosody (details in Section 4.4). The optimization goal of the prosody predictor is to minimize the KL divergence between the predicted normal distribution and the posterior distribution. Therefore, the training loss of the prosody predictor is as follows:

$$\mathcal{L}_{pp} = D_{kl}(\mathcal{N}(\mu_{pp}, \sigma_{pp}), \mathcal{N}(\mu_{\phi}, \sigma_{\phi})), \tag{6}$$

where $\mu_{pp}$ and $\sigma_{pp}$ are the mean and standard deviation predicted by the prosody predictor. Compared with Fastspeech 2 to directly predict pitch and energy, we predict the distribution of phoneme-level prosody, avoiding the one-to-many mapping problem.

It is worth noting that the duration predictor we used is an improved version of the duration predictor in FastSpeech 2 [26]. Since prosodic information partly determines the phoneme duration, we use $z$ as the condition of the duration predictor besides the output of the text encoder, i.e., $\hat{\mathbf{d}} = \mathcal{DP}(z, t) \in \mathbb{R}^{T_p}$, where $\mathcal{DP}$ denotes the duration predictor, and $t$ is the output of the text encoder. In this way, CLONE can generate more stable and natural phoneme durations.

### 3.4 Waveform Generation

#### 3.4.1 Motivation

Some end-to-end TTS studies [26, 7, 15] focus on generating raw waveforms directly from phonemes recently. These single-stage TTS systems usually generate learned intermediate representations from phonemes and then synthesize raw waveforms from the learned intermediate representations. Unlike the mel-spectrogram used in two-stage methods, the intermediate representation in single-stage methods is predicted by the model without training supervision, subject to prediction errors and over-smoothness. The lack of supervision of intermediate representations during training expands the search space of the single-stage model, resulting in the model being more challenging to optimize, which is reflected in the poor modeling ability for high-frequency information in our experiments. To narrow the search space of waveform generation in single-stage models, we introduce ground-truth speech signals. We design an autoencoder architecture called Dual Parallel Autoencoder (DPA) to regularize the learned intermediate representation. In addition, to further enhance the quality of the generated waveforms, we propose a new discriminator called Multi-Band Discriminator (MBD). MBD divides the waveform into multiple bands so that our model can separately supervise the low-frequency and high-frequency parts of the audio, improving the overall quality of the synthesized speech.

#### 3.4.2 Dual Parallel Autoencoder

In the dual parallel autoencoder, two parallel encoders, namely the acoustic encoder and the posterior wave encoder, generate intermediate representation, and a dual training method is used to optimize them. The acoustic encoder transforms $z$ into the predicted intermediate representations $\hat{ir}$ conditioning on linguistic features, and the posterior wave encoder transforms linear spectrograms into the intermediate representations $ir$. The wave decoder acts as the decoder of DPA and generates the waveform from both intermediate representations. In practice, we concatenate $ir$ and $\hat{ir}$ in the batch dimension to get $ir_{concat}$ and send $ir_{concat}$ into the wave decoder to produce the waveform $\hat{w}$. For dual training, we compute the mean absolute error (MAE) $\mathcal{L}_{ir}$ between $ir$ and $\hat{ir}$, so that $\hat{ir}$ gets the supervision of ground-truth acoustic features from $ir$, and $\hat{ir}$ is regularized by $ir$, which assists CLONE in learning intermediate representation efficiently. Please note that we only use the acoustic encoder without the posterior wave encoder during inference.

To calculate the reconstruction loss, we convert $\hat{w}$ to mel-spectrogram $m_1$ and calculate MAE with ground-truth mel-spectrogram $m_{gt}$. To make $ir$ and $\hat{ir}$ focus on the information at the human voice frequency band for better prosody modeling, we use a one-layer linear network to predict the mel-spectrogram $m_2$ from $ir_{concat}$. Therefore, the whole reconstruction loss $\mathcal{L}_{recon}$ is:

$$\mathcal{L}_{recon} = \text{MAE}(m_{gt}, m_1) + \text{MAE}(m_{gt}, m_2). \tag{7}$$

### 3.4.3 Discriminator

We use the popular adversarial training approach as HiFi-GAN [17] to improve the resolution of synthesized speech. The following equations describe the loss of adversarial training:

$$\mathcal{L}_{advD} = \mathbb{E}_{(w,\hat{w})} \left[ ((D(w) - 1)^2 + (D(\hat{w}))^2 \right], \tag{8}$$

$$\mathcal{L}_{advG} = \mathbb{E}_w \left[ (D(\hat{w}) - 1)^2 \right] + \beta\mathcal{L}_{fm}, \quad \mathcal{L}_{fm} = \mathbb{E}_{(w,\hat{w})} \left[ \sum_{l=1}^{L} \frac{1}{N_l} \mathrm{MAE}(D_l(w), D_l(\hat{w})) \right], \quad (9)$$

where $\mathcal{L}_{advD}$, $\mathcal{L}_{advG}$ and $\mathcal{L}_{fm}$ denote the loss of the discriminator, the loss of the wave decoder, and the loss of the feature map, respectively. $L$ denotes the total number of layers in the discriminators. $D_l$ and $N_l$ denote the features and the number of features in the $l$-th layer of the discriminator, respectively. $\beta$ is the coefficient of the feature map loss $\mathcal{L}_{fm}$, and we set it to 0.1 in our experiments.

We use two discriminators for joint adversarial training, namely MPD in HiFi-GAN and MBD. MBD uses the pseudo quadrature mirror filter bank (Pseudo-QMF) to divide the waveform into $N$ sub-bands. These $N$ sub-bands with one full-band are respectively sent to $N + 1$ scale discriminators in MelGAN [18]. By applying different discriminators on different frequency bands of the synthesized audio, MBD can significantly enhance the generation quality of high-frequency parts, allowing CLONE to generate high-fidelity and lossless audio at a high sample rate. A similar idea is used in [21]. However, our method differs in that we send different frequency bands into different discriminators.

### 3.5 Loss Function

The total loss of CLONE can be expressed as follows:

$$\mathcal{L} = \lambda_1 * \mathcal{L}_{recon} + \lambda_2 * \mathcal{L}_{kl} + \lambda_3 * \mathcal{L}_{ir} + \lambda_4 * \mathcal{L}_{pp} + \lambda_5 * \mathcal{L}_{dur} + \lambda_6 * \mathcal{L}_{advG}, \tag{10}$$

where $\lambda_{[1 \to 6]}$ represent coefficients of different components of the total loss.

## 4 Experiment

### 4.1 Dataset

We used proprietary English speech datasets, including a single-speaker dataset and a multi-speaker dataset. The single-speaker dataset contains 11,176 utterances with a total audio length of 10 hours at both 24 kHz and 48 kHz. We use 9,000 utterances as the training set, 100 utterances as the validation set, and the remaining data as the test set. The multi-speaker speech data contains five English speakers (two males and three females) with a total audio length of 22 hours. We evaluate the high-quality generation capability of CLONE on the single-speaker dataset and the prosody transfer capability of CLONE on the multi-speaker dataset.

### 4.2 Data Preprocessing

We convert the text sequences into phoneme sequences following [2, 19, 26]. In experiments, we use 80-dimensional mel-spectrograms and linear spectrograms with 2048 filters. For the audio at 24 kHz, the hop size is 300, and the window size is 1200. For the audio at 48 kHz, the hop size is 300, and the window size is 2048. All use the Hann window.

### 4.3 Model Configuration

Our text encoder uses a stack of 6 feed-forward transformer blocks [35], and the prosody predictor consists of 4 WaveNet residual blocks (dilated CNN) [22], which consists of layers of dilated convolutions with a gated activation unit and skip connection. The duration predictor consists of a 2-layer 1D convolutional network with ReLU activation, each followed by layer normalization [3] and the dropout layer [31], and an extra linear layer with ReLU activation to output a scalar, which is

the predicted phoneme duration. The posterior encoder consists of 8 blocks of dilated CNN, and the normalizing flow is a stack of affine coupling layers [6] consisting of 4 blocks of dilated CNN. The acoustic encoder is composed of 8 blocks of dilated CNN. The posterior wave encoder consists of 8 blocks of dilated CNN. The structure of the wave decoder is the same as HiFi-GAN V1. To match our hop size, we change the upsampling rate to 5, 5, 4, 3 and change the upsampling kernel size to 15, 15, 12, 9. The detailed hyper-parameters of CLONE are listed in the appendix.

## 4.4 Training and Inference

We train our model on 4 NVIDIA Tesla V100 32G GPUs with the batch size of 16 on each GPU for 500k steps. We use the AdamW [20] optimizer with $\beta_1 = 0.8$ and $\beta_2 = 0.99$. The learning rate of CLONE is fixed at 2e-4. The discriminator uses the same optimization settings, with a fixed learning rate of 1e-4. As VITS, to reduce training time and GPU memory usage, we employ a windowed generator for training, randomly sampling segments of intermediate representation with a window size of 32 frames as input to the wave decoder.

CLONE has three modes of inference, and the difference lies in how the prosody latent variable is calculated. (a) Use the prosody predictor to predict prosody information based on text information, and the input of the model is only text information, just like a typical TTS model. (b) The prosody latent variable is obtained through the inverse flow transformation, where the inputs of CLONE are textual information and the sampling value of the standard normal distribution. (c) The prosody information is extracted by the posterior encoder from the input linear spectrogram. In this mode, the model inputs are text information and the reference linear spectrogram. Besides, we test the inference speed of CLONE in mode (a) on an NVIDIA Tesla V100 32G GPU and compare it to that of VITS. The RTF of CLONE and VITS are 0.012 and 0.017, respectively, indicating that our model has a comparable inference speed to the SOTA single-stage parallel TTS model.

## 4.5 Evaluation

### 4.5.1 MOS Evaluation

We conduct the MOS (mean opinion score) evaluation to measure the synthesis quality of different models. We use each model to synthesize the same 30 utterances[1], and let 15 English native speakers evaluate them. We compare with the state-of-the-art (SOTA) autoregressive TTS model Tacotron 2 [29] with GMM-based attention mechanisms [4], the SOTA non-autoregressive TTS model FastSpeech 2 [26], and the SOTA single-stage TTS model VITS [15]. The vocoder for Tacotron 2 and FastSpeech 2 is TFGAN [33], as TFGAN has better generation robustness than HiFi-GAN in our experiments. The above models all generate audio at 24 kHz. We evaluate two kinds of CLONE, namely 24 kHz CLONE and 24 kHz CLONE without MBD, of which the discriminators are the same as those used in HiFi-GAN. The results of the MOS evaluation are shown in Table 1. It can be seen that CLONE surpasses other SOTA models, indicating that our phoneme-level prosody modeling method and the introduction of DPA enable the model to synthesize highly natural speech with appropriate prosodic inflections. In addition, CLONE is better than CLONE without MBD, demonstrating the effectiveness of MBD for generating higher quality speech.

Table 1: The MOS result with $95\%$ confidence intervals of different models.

| Method | MOS | CI |
|---|---|---|
| Tacotron 2 + TFGAN | 4.0717 | $\pm 0.0578$ |
| FastSpeech 2 + TFGAN | 4.0983 | $\pm 0.0579$ |
| VITS | 4.0108 | $\pm 0.0601$ |
| 24 kHz CLONE w/o MBD | 4.1000 | $\pm 0.0606$ |
| **24 kHz CLONE** | **4.1567** | $\pm 0.0528$ |

---

[1]To test the synthesis quality and robustness of the model and avoid data leakage, we use 30 general utterances outside the dataset for testing.

Furthermore, we conduct CMOS evaluations on 30 utterances with ground-truth recording audio. We compare 24 kHz VITS, 24 kHz CLONE, and 48 kHz CLONE with 48 kHz ground-truth audio, respectively, as shown in Table 2. We find that 48 kHz CLONE can generate the audio close to the 48 kHz ground-truth audio and outperform both the 24 kHz CLONE and 24 kHz VITS, demonstrating that CLONE can synthesize high-fidelity 48 kHz audio.

Table 2: The CMOS comparison to evaluate speech generation quality at high sample rates. The audio synthesized by 48 kHz CLONE, 24 kHz CLONE, and 24 kHz VITS is compared with 48 kHz ground-truth audio, respectively.

| Method | CMOS |
|---|---|
| 48 kHz Ground-truth Audio | 0 |
| 48 kHz CLONE | $-\mathbf{0.1712}$ |
| 24 kHz CLONE | $-0.2393$ |
| 24 kHz VITS | $-0.3621$ |

### 4.5.2 Prosody Modeling

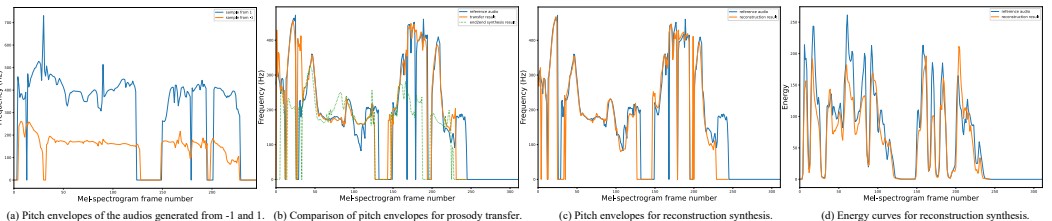

(a) Pitch envelopes of the audios generated from -1 and 1.    (b) Comparison of pitch envelopes for prosody transfer.    (c) Pitch envelopes for reconstruction synthesis.    (d) Energy curves for reconstruction synthesis.

Figure 2: Visualization of prosody modeling. (a) shows the variation in the pitch of the audio generated from all $-1$ and all $1$ sampling, respectively. (b) shows the pitch comparison of the reference audio, transfer result, and end-to-end synthesis result. (c) and (d) show the pitch and energy of the reconstruction result and reference audio, respectively.

**Prosody Variation**    We infer CLONE in mode (b), i.e., sampling values in the standard normal distribution and obtaining the phoneme-level prosody latent variable through the inverse flow transformation. We find that the prosody of the generated speech has a very high variance by sampling different values in the standard normal distribution, which further proves the prosody modeling capability of CLONE. Figure 2a visualizes the variation in the pitch of the audio generated by setting sample values to all $-1$ and all $1$, respectively. It can be seen that a significant prosody variation can be achieved by adjusting sample values in the standard normal distribution of the inverse flow transformation.

**Prosody Transfer**    We test the prosody transfer performance of CLONE. Firstly, we train a multi-speaker CLONE [2]. Then we infer CLONE in mode (c). We input the reference speech of speaker 1 to the posterior encoder and use the speaker embedding of speaker 2 and the duration of the reference speech to synthesize the transfer result. The goal is to use the timbre of speaker 2 to synthesize audio with the same prosody as the reference audio of speaker 1. We also use the end-to-end synthesis result with the timbre of speaker 2 for comparison. In Figure 2b, the pitch envelope of the transfer result is very similar to that of the reference audio and quite different from that of the end-to-end synthesis result. Besides, we calculate the MAE on pitch and energy of the prosody transfer result and the end-to-end synthesis result (both are synthesized using the duration of the reference audio) with the reference audio as ground truth, as shown in Table 3. It can be seen that the MAE of the prosody transfer result is significantly smaller than that of the end-to-end synthesis result. Above two experiments prove the effectiveness of prosody transfer.

---

[2]To implement multi-speaker TTS, we add speaker embedding to prosody predictor, duration predictor, posterior encoder, and acoustic encoder.

Table 3: The mean absolute error of pitch and energy of different methods.

| Method | Pitch MAE | Energy MAE |
|---|---|---|
| end-to-end synthesis | 79.38 | 35.91 |
| prosody transfer synthesis | **41.44** | **31.92** |
| reconstruction synthesis by CLONE with HAPE | **32.56** | **16.34** |
| reconstruction synthesis by CLONE with SAPE | 64.83 | 23.70 |

**Prosody Reconstruction**  CLONE can extract the prosody of the reference audio through the posterior encoder and then reconstruct the audio as a conditional VAE. We draw the pitch and energy curves of the reference audio and the reconstruction result, as shown in Figure 2c and Figure 2d. We find that the two curves in each figure are very similar, indicating that CLONE can accurately extract and reconstruct the prosody (e.g., pitch and energy) of the reference audio.

### 4.5.3  Ablation Study

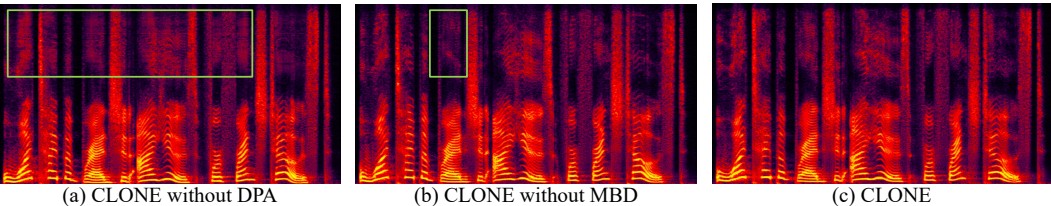

(a) CLONE without DPA          (b) CLONE without MBD          (c) CLONE

Figure 3: The spectrograms generated by three different CLONE, (a) CLONE without DPA, (b) CLONE without MBD, and (c) complete CLONE.

**Waveform Generation**  To further investigate the improvement of our method on waveform generation, we conduct ablation experiments. Firstly, we conduct a MOS evaluation on the audio synthesized by the CLONE with and without MBD, as shown in Table 1. We find that MBD enhances the quality of synthesized audio. In addition, we plot the spectrograms of the synthesized audio, as shown in Figure 3. Figure 3a shows that without the DPA, the synthesized audio suffers obvious over-smoothness at high frequency, and Figure 3b shows that without MBD, the high-frequency details of synthesized audio are insufficient. These demonstrate that DPA significantly weakens the over-smoothness of the high frequency, and MBD further enhances the high-frequency details.

**Prosody Extraction**  CLONE uses a hard alignment prosody extractor (HAPE) to extract prosody, which improves the accuracy of prosody extraction. To verify this, we compare HAPE with the soft attention prosody extractor (SAPE) [9, 32] where the text encoder output is to query the ground-truth linear spectrogram by soft attention. We compute the MAE on pitch and energy of the above two methods in Table 3. It can be seen that the MAE of HAPE is smaller than that of SAPE, indicating that HAPE can extract prosody more accurately.

## 5  Conclusion

In this work, we propose a single-stage TTS system called CLONE that can directly generate lossless waveforms from the text in parallel. Specifically, we design a phoneme-level prosody modeling method based on a variational autoencoder with normalizing flows and a prosody predictor to solve the one-to-many mapping problem better and support end-to-end expressive speech synthesis. Besides, the dual parallel autoencoder is introduced to solve the problem of lacking supervision of ground-truth acoustic features during training, which allows the single-stage model to generate lossless speech. Our experiments demonstrate that CLONE outperforms existing SOTA single-stage and two-stage TTS models in speech quality while performing strong controllability over prosody.

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
