# OpenReview forum: "Controllable and Lossless Non-Autoregressive End-to-End Text-to-Speech"
_NeurIPS.cc/2022/Conference — NeurIPS 2022 Submitted_

### Official Review · Reviewer_xNqf · 2022-07-11

**Rating:** 3
**Confidence:** 4
**Soundness:** 1 poor
**Presentation:** 1 poor
**Contribution:** 1 poor

**Summary:**

This paper introduces a text-to-speech model that combines several generative models such as VAE, GAN, and Normalizing Flow. The authors propose to use phoneme-level latent variable to generate prosodic latent variable. Normalizing Flow was used to make more complex probability distributions for the latent space of VAE. The evaluation on internal dataset shows that the proposed method works better than recent end-to-end SOTA model, VITS.

**Questions:**

- line 121, “We use the obtained duration of phonemes to construct the hard alignment matrix” → The process of obtaining the duration of phonemes was not mentioned before this.
- How do the authors safely assume that the ouput of prosody predictor is actually prosodic latent variable?
- Why do text information included in both reference and source text the same? This is not a practical scenario and not a fair way to test the actual performance.
- Why is approximate posterior distribution denoted as q(z|x,c)? Isn’t it q(z|x)?
- It seems not technically correct to call the transformed prior with normalizing flow “a conditional prior (p(z|c)) because no condition was actually given. It makes sense in the paper VITS because the conditional prior distribution is constructed by feeding a text information to the network.
- Why do the authors use TFGAN for fastspeech2 and tacotron2 baselines when there’s HiFi-GAN vocoder that is used very often in many recent works? I don’t think I have seen TTS works that use TFGAN for the vocoder.
- Why do the authors evaluate the proposed model using internal datasets?

**Limitations:**

Both the limitations and potential negative societal impact are written very briefly in Appendix.

**Strengths And Weaknesses:**

- strengths
    - The authors try to disentangle latent information that was entangled in a single random variable in VITS.

- weakness
    - more thorough subjective evaluations are required to evaluate how good the prosody modeling of the proposed method is. Simple MOS is not enough.
    - Some technical details seem wrong. See Questions.

---

> ### Author Response · Authors · 2022-08-02
> **Reply to Reviewer xNqf (1)**
>
> Thanks for your comments on our paper. We reply to your questions as follows:
>
> > line 121, "We use the obtained duration of phonemes to construct the hard alignment matrix" → The process of obtaining the duration of phonemes was not mentioned before this.
>
> During training, we get the hard alignment matrix $\textbf{A}$ from the ground-truth durations $\mathbf{d} = \{d_1, ..., d_{T_p}\}$ of phonemes. $\mathbf{A} \in \mathbb{R}^{T_p \times T_f}$ and each element in A is 0 or 1.
>
> $A_{ij} = 1$, when $\sum_{k=1}^{i-1}d_k \leq j < \sum_{k=1}^{i}d_k$
>
> $A_{ij} = 0$, when otherwise.
>
> In other words, $A_{ij} = 1$ denotes The i-th phoneme is being pronounced at the j-th spectrogram frame.
>
> > How do the authors safely assume that the output of prosody predictor is actually prosody latent variable?
>
> 1. Firstly, the learning target of the prosody predictor is the distribution of phoneme-level prosody generated by the posterior encoder. We assume that the posterior encoder can extracts the phoneme-level prosody latent variable.
> 2. Secondly, in order to allow the posterior encoder to accurately extract the prosody information and filter out other information, such as linguistic features, we conduct an average pooling operation to downsample on the frame-level prosody features. Average pooling eliminates timing information between phones and ensures that linguistic information is removed.
> 3. Our experiments demonstrate the effectiveness of our method. In  prosody reconstruction, the pitch and energy of the generated audio are extremely similar to the reference audio. By inputting different sample values to invertible transformation of normalizing flow, the generated audio achieves high variation of pitch, while linguistic information is not affected.
>
> > Why is text information included in both reference and source text the same? This is not a practical scenario and not a fair way to test the actual performance.
>
> We performed prosody transfer and prosody reconstruction experiments to demonstrate the capability of CLONE's prosody modeling. This capability is divided into two aspects, one is to accurately extract the prosody, and the other is that the extracted prosody is independent with the text and speaker, and can be transferred between different speakers and different texts. For the first aspect, the prosody reconstruction experiment shows that CLONE can accurately extract the prosody, and the extracted prosody is very similar to the prosody of the reference audio. For the second aspect, our prosody transfer experiment proves that prosody can be transferred between speakers, and the prosody variation experiment proves that prosody is independent with text.
>
> > Why is approximate posterior distribution denoted as q(z|x,c)? Isn't it q(z|x)?
>
> Here we use condition VAE to design our model. Referring to the theory of condition VAE, we introduce the condition linguistic feature in the posterior encoder and prior encoder, which is $c$, so in CLONE, the posterior encoder is $q_{\phi}(z \mid x, c)$. To understand intuitively, we provide the posterior encoder with linguistic features, which help the posterior encoder filter out linguistic information from the input linear spectrogram and model the prosody.
>
> > It seems not technically correct to call the transformed prior with normalizing flow a conditional prior (p(z|c)) because no condition was actually given. It makes sense in the paper VITS because the conditional prior distribution is constructed by feeding text information to the network.
>
> There is no technical error here. First of all, our assumption is similar to variational inference with normalizing flow[1] and VITS[2]. The posterior encoder outputs a normal distribution, which is then transformed by the normalizing flow to another normal distribution. Unlike VITS, the normalizing flow in VITS transforms the input distribution to the non-standard normal distribution predicted by the text encoder, and we transform the input distribution to the standard normal distribution. Besides, $z$ is latent prosody variable, which is independent with linguistic information $c$, thus, $p_{\theta}(z \mid c) = p_{\theta}(z)$. Thus, we have $p_{prior} = p_{\theta}(z \mid c) = p_{\theta}(z) = \mathcal{N}\left(f_{\theta}(z) ; \mathbf{0}, \boldsymbol{I}\right) \left|\operatorname{det} \frac{\partial f_{\theta}(z)}{\partial z}\right|$.
>
> - [1] Variational Inference with Normalizing Flows.
> - [2] Conditional Variational Autoencoder with Adversarial Learning for End-to-End Text-to-Speech.

---

> ### Author Response · Authors · 2022-08-02
> **Reply to Reviewer xNqf (2)**
>
> > Why do the authors use TFGAN for fastspeech2 and tacotron2 baselines when there's HiFi-GAN vocoder that is used very often in many recent works? I don't think I have seen TTS works that use TFGAN for the vocoder.
>
> After evaluation, our TFGAN performs better than the open source HiFi-GAN V1. Our TFGAN is able to generate higher quality waveform with sharper high frequency and less glitch. For a fairer comparison and to show the high quality of the waveform generated by CLONE, we chose TFGAN.
>
> > Why do the authors evaluate the proposed model using internal datasets?
>
> We used an internal dataset because the internal dataset has the following advantages:
> 1. Firstly, our internal data has rich prosody diversity, which puts forward higher requirements for the prosody modeling capability of the model.
> 2. Secondly, our internal dataset has high-quality recordings and 48k audio, which can fully demonstrate the model's capability of waveform generation.
>
> ### Summary
>
> In particular, our paper proposes a new prosody modeling method. We model the phoneme-level prosody in an implicit way, assuming that the prosody obeys a non-standard normal distribution. It cannot fully characterize prosody information to use pitch or energy to model prosody explicitly such as FastSpeech 2, FastPitch and etc. In addition, we use the prosody predictor to predict the prosody distribution instead of the fixed prosody vector, which can avoid the over-smoothness of the prosody prediction, because for the same text, there will be different prosody distributions. Therefore, we assume that prosody obeys a non-standard normal distribution, which increases the upper bound of prosody representational power, and CLONE's prosody predictor avoids the over-smoothness of prosody prediction, improving the expressiveness of the generated audio in end-to-end synthesis. CLONE can achieve rich prosody control capabilities, in addition to those mentioned in the paper, CLONE can use the timbre of speaker A to synthesize the speech of speaker B's speaking style by replacing the speaker ID of the prosody predictor with speaker B's. Samples are at [here](https://github.com/CLONEneurips2022/samples).
>
> Besides, we introduce the dual parallel autoencoder to solve the problem of lacking supervision of ground-truth acoustic features during training, which allows the single-stage model to generate higher quality speech.
>
> We hope our replies solve all your concerns about the paper. If we successfully address your concerns, we would strongly appreciate an increased score; otherwise, we are happy to provide additional discussion and address any further questions. Thanks!

---

> ### Author Response · Authors · 2022-08-08
> **Reply to Reviewer xNqf**
>
> Dear Reviewer xNqf:
>
> We thank you for the precious review time and valuable comments. We have provided corresponding responses and results, which we believe have covered your concerns. We hope to further discuss with you whether or not your concerns have been addressed. Please let us know if you still have any unclear parts of our work.
>
> Best.

---

### Official Review · Reviewer_1KcR · 2022-07-12

**Rating:** 3
**Confidence:** 5
**Soundness:** 1 poor
**Presentation:** 1 poor
**Contribution:** 1 poor

**Summary:**

The authors propose a controllable and lossless method of Non-Autoregressive End-to-End Text-to-Speech synthesis.

**Questions:**

Included in what was described above.

**Limitations:**

The authors mentioned limitations in the appendix.

**Strengths And Weaknesses:**

* Weaknesses
1. The authors claim that the proposed method generates lossless speech, but comparing the human recording and synthesized audio presented by the authors clearly shows that there is a quality loss. Also, considering that it is a generative model based on neural networks, there is an error in the authors' claim.
1. The authors claim that the proposed method achieves high controllability, but I would like to ask what can be controlled. There is no evidence in the paper that the desired speech audio can be synthesized by controlling the model proposed in this work.
1. The authors claim the following:
”Unlike the mel-spectrogram used in two-stage methods, the intermediate representation in single-stage methods is predicted by the model without training supervision, subject to prediction errors and over-smoothness.”\
It is not clear which intermediate features in single stage models are claimed, and the authors should provide scientific analysis of the problem of intermediate features pointed out by the authors.
1. The authors claim the following:
“we carefully study the problem of unsatisfactory high-frequency information generation in single-stage end-to-end speech synthesis, which is caused by the lack of the supervision of ground truth acoustic features during training.”\
Considering that other models are also supervised using mel-spectrogram loss similar to the proposed method, it is difficult to see that other models lack the supervision of the ground truth, and the method proposed is fundamentally the same as the previous work in the point of view.
1. It is necessary to clarify whether the inability to synthesize high frequencies with high quality is a common problem that occurs with other models or a problem that only occurs with the proposed model. In addition, in order to claim that DPA is a general method to increase the quality of high frequency bands, it must be shown that it is effective even when applied to other models.
1. The authors did not specify how the comparative models were trained. Referring to the quality of audio samples, the performance of some models differs in quality from the audio samples presented in the original work or best-performing open source implementations. Given that the human recording samples presented by the authors show uniform quality than the widely used public dataset, some models appear to be compared without convergence. All comparisons should be made between sufficiently converged models, and the authors should specify the training settings of the comparison models, and if they did not converge to the level suggested in the original work, the authors should re-experiment and revise the comparison results.
1. Most recent work use widely used public datasets and author's official implementations and pre-trained weights for fair and unambiguous comparisons. The models used by the authors for comparison are models with well-performing public implementations and pre-trained weights. By training the proposed model with the public dataset(eg, LJSpeech), authors can easily compare it with other models. This is the fairest and most reliable comparison method that can claim that the proposed method shows state-of-the-art performance.
1. Some figures need to be improved, such as font size.

---

> ### Author Response · Authors · 2022-08-02
> **Reply to Reviewer 1KcR (1)**
>
> Thanks for your comments on our paper. We reply to your questions as follows:
>
> > The authors claim that the proposed method generates lossless speech, but comparing the human recording and synthesized audio presented by the authors clearly shows that there is a quality loss. Also, considering that it is a generative model based on neural networks, there is an error in the authors' claim.
>
> 48kHz CLONE can generate audio with 48k sample rate and 16 bit depth. Generally speaking,  48kHz and 16 bit audio is "lossless audio", so we consider CLONE can synthesize lossless audio. At the same time, we also admit that there is a gap between the audio generated by 48kHz CLONE and ground-truth 48kHz audio. It should be emphasized that CLONE is the first single-stage TTS model capable of generating 48kHz audio and we show the possibility of a single-stage TTS model for generating higher sample rate audio.
>
> > The authors claim that the proposed method achieves high controllability, but I would like to ask what can be controlled. There is no evidence in the paper that the desired speech audio can be synthesized by controlling the model proposed in this work.
>
> CLONE can achieve control of prosody through the following methods:
> 1. Sample different value from standard normal distribution as the input of normalizing flow to achieve control of generated audio. As shown in Figure 2(b), the input of flow is all set to -1 or 1 and the generated audio has high pitch variation. You can also refer to the demo page in the [section of "Prosody Variation"](https://cloneneurips2022.github.io/CLONE/). The variation of generated speech by distributing different sampling values has been shown in many previous works, like Flowtron[1], Parallel Tacotron and Glow-TTS[2], and these works also claim to achieve controllability.
> 2. CLONE can use speaker A's audio as reference audio to generate speaker B's audio with speaker A's prosody. The samples are shown on the demo page in the section of "Prosody Transfer" and we analysis the pitch and energy similarity in Figure 2 (c) & (d) and Table 3. Unlike some works (FastSpeech 2[3], FastPitch[4], etc) which explicitly model pitch and energy, this experiment proves that our method can implicitly model prosody to achieve the transfer of pitch and energy.
> 3. CLONE can use the timbre of speaker A to synthesize the speech of speaker B's speaking style by replacing the speaker ID of the prosody predictor with speaker B's. Samples are at [here](https://github.com/CLONEneurips2022/samples).
>
> > The authors claim the following: "Unlike the mel-spectrogram used in two-stage methods, the intermediate representation in single-stage methods is predicted by the model without training supervision, subject to prediction errors and over-smoothness."
> It is not clear which intermediate features in single stage models are claimed, and the authors should provide scientific analysis of the problem of intermediate features pointed out by the authors.
>
> In most previous two-stage neural network TTS models, the mel spectrogram is first predicted by the acoustic model, and then the mel spectrogram is converted into a raw waveform by the vocoder. There is no such process in the single-stage TTS model, but most of the single-stage TTS models maintain a similar two-stage mode, which first extends the input text information to the intermediate representation with the spectrogram length, and then predicts the waveform from the intermediate representation by a module similar to vocoder, such as FastSpeech 2s, VITS, EATS and etc.
>
> In the two-stage neural network TTS model, the predicted mel spectrogram usually has prediction error and over-smoothness, and these problems weaken the vocoder to generate high-quality waveforms. Thus, we need to use ground-truth mel spectrogram to train the vocoder, and during fine tuning, we use gta mel spectrogram to fine tune the vocoder, similar to Tacotron and Tacotron2. Just as the two-stage model predicts the mel spectrogram with prediction error and over-smoothness, the single-stage model also has the same problem when predicting intermediate representations. These predicted intermediate representations are with over-smoothness and not completely aligned with the raw waveform. These problems will also slow down the convergence of the second part (intermediate representation to waveform) of the single-stage model and weaken the generated audio quality. Specifically, these problems will expand the search space of the second part of the single-stage model and produce errors in the mapping relationship between the intermediate representation and waveform.
>
> - [1] Flowtron: an Autoregressive Flow-based Generative Network for Text-to-Speech Synthesis.
> - [2] Glow-TTS: A Generative Flow for Text-to-Speech via Monotonic Alignment Search.
> - [3] FastSpeech 2: Fast and High-Quality End-to-End Text to Speech.
> - [4] FastPitch: Parallel Text-to-speech with Pitch Prediction

---

> ### Author Response · Authors · 2022-08-02
> **Reply to Reviewer 1KcR (2)**
>
> > The authors claim the following: "Unlike the mel-spectrogram used in two-stage methods, the intermediate representation in single-stage methods is predicted by the model without training supervision, subject to prediction errors and over-smoothness."
> It is not clear which intermediate features in single stage models are claimed, and the authors should provide scientific analysis of the problem of intermediate features pointed out by the authors.
>
> In most previous two-stage neural network TTS models, the mel spectrogram is first predicted by the acoustic model, and then the mel spectrogram is converted into a raw waveform by the vocoder. There is no such process in the single-stage TTS model, but most of the single-stage TTS models maintain a similar two-stage mode, which first extends the input text information to the intermediate representation with the spectrogram length, and then predicts the waveform from the intermediate representation by a module similar to vocoder, such as FastSpeech 2s, VITS[5], EATS[6] and etc.
>
> In the two-stage neural network TTS model, the predicted mel spectrogram usually has prediction error and over-smoothness, and these problems weaken the vocoder to generate high-quality waveforms. Thus, we need to use ground-truth mel spectrogram to train the vocoder, and during fine tuning, we use gta mel spectrogram to fine tune the vocoder, similar to Tacotron and Tacotron2. Just as the two-stage model predicts the mel spectrogram with prediction error and over-smoothness, the single-stage model also has the same problem when predicting intermediate representations. These predicted intermediate representations are with over-smoothness and not completely aligned with the raw waveform. These problems will also slow down the convergence of the second part (intermediate representation to waveform) of the single-stage model and weaken the generated audio quality. Specifically, these problems will expand the search space of the second part of the single-stage model and produce errors in the mapping relationship between the intermediate representation and waveform.
>
> > The authors claim the following: "we carefully study the problem of unsatisfactory high-frequency information generation in single-stage end-to-end speech synthesis, which is caused by the lack of supervision of ground truth acoustic features during training."
> Considering that other models are also supervised using mel-spectrogram loss similar to the proposed method, it is difficult to see that other models lack the supervision of the ground truth, and the method proposed is fundamentally the same as the previous work in the point of view.
>
> The "lack of supervision" mentioned here does not mean the lack of supervision from loss function, but the lack of supervision in the model search space during the training process. As above, the prediction error and over-smoothness of the intermediate representation will affect the convergence of the second part of single-stage model, which increases the search space of the second part and weakens the quality of generated audio. In the two-stage model, the vocoder has ground-truth mel spectrogram as input in the training phase, and learns the mapping relationship between the ground-turth mel spectrogram and the raw waveform, whose search space is smaller than the former. In the single-stage model, although the intermediate representation has mel spectrogram loss as the loss function, the prediction error and over-smoothness always exist. We need "supervision" of the ground-truth signal, so we designed the posterior wave encoder. This module takes the ground-truth linear spectrogram as input, and the output of this module is the input of the second part of single-stage model. In the pipeline of "linear spectrogram -> parallel wave encoder -> intermediate representation -> wave generator -> waveform", the model learns the mapping relationship between linear spectrogram and waveform. Compared with the initial, this method narrows the search space and speeds up convergence. In general, we introduce the ground-truth signal into the second part in the form of a parallel encoder, thereby reducing the search space of the second part, and finally improving the generated audio quality.
>
> - [5] Conditional Variational Autoencoder with Adversarial Learning for End-to-End Text-to-Speech.
> - [6] End-to-End Adversarial Text-to-Speech.

---

> ### Author Response · Authors · 2022-08-02
> **Reply to Reviewer 1KcR (3)**
>
> > It is necessary to clarify whether the inability to synthesize high frequencies with high quality is a common problem that occurs with other models or a problem that only occurs with the proposed model. In addition, in order to claim that DPA is a general method to increase the quality of high frequency bands, it must be shown that it is effective even when applied to other models.
>
> 1. The high frequency generation problem also exists in other single-stage models, such as FastSpeech 2s and EATS. The fundamental reason is that there is a strong mapping relationship between textual information and vocal frequency band information (such as mel spectrogram), but the mapping relationship between textual information and high-frequency information is not significant. Mel spectrogram loss will constrain the model to reconstruct human voice information, but it is difficult to constrain the model learn high frequencies. As mentioned above, the prediction error and over-smoothness of the intermediate representation will affect the convergence of the second part of single-stage model, which increases the search space of the second part and weakens the quality of generated audio. Compared with the information of vocal frequency band that is easy to learn, high-frequency information is difficult to model efficiently, which is manifested as poor high-frequency generation ability of the single-stage model. It is worth noting that VITS can solve this problem by introducing a VAE to generate waveform from linear spectrogram, and we have introduced a new idea to solve this problem.
> 2. The architecture of DPA can be used in other single-stage models to improve high-frequency modeling capabilities. Single-stage model can design a posterior wave encoder with linear spectrogram as input and intermediate representation as output. Calculate loss with intermediate representation generated by the first part of single-stage model, and jointly optimize, to narrow the search space of second part, and improve the quality of generated audio.
>
> In general, our motivation for designing the DPA structure is that we found that the pipeline of "text->acoustic model->spectrogram-length intermediate representation->upsampler->waveform" cannot generate high-quality waveforms, and the fundamental reason is that the quality of the "spectrogram-length intermediate representation" is poor, which slows down the convergence of the upsampler and affects the high-frequency reconstruction of the upsampler. So we design the posterior wave encoder to introduce the information from the ground-truth linear spectrogram into the training process of the "upsampler" and provide more supervision to the "spectrogram-length intermediate representation".
>
> > The authors did not specify how the comparative models were trained. Referring to the quality of audio samples, the performance of some models differs in quality from the audio samples presented in the original work or best-performing open source implementations. Given that the human recording samples presented by the authors show uniform quality than the widely used public dataset, some models appear to be compared without convergence. All comparisons should be made between sufficiently converged models, and the authors should specify the training settings of the comparison models, and if they did not converge to the level suggested in the original work, the authors should re-experiment and revise the comparison results.
>
> The comparative models we provide, VITS, FastSpeech 2 and Tacotron 2, are all open source implementations, trained using our internal data. TFGAN[7] is an internal implementation. After evaluation, our TFGAN's performance is stronger than the open source HiFi-GAN V1. For the fairness of the comparison, we specially selected the checkpoint with the best degree of convergence. Thus, we believe that the comparative models we provide can be compared with CLONE fairly.
>
> - [7] TFGAN: Time and Frequency Domain Based Generative Adversarial Network for High-fidelity Speech Synthesis.

---

> ### Author Response · Authors · 2022-08-02
> **Reply to Reviewer 1KcR (4)**
>
> > Most recent work use widely used public datasets and author's official implementations and pre-trained weights for fair and unambiguous comparisons. The models used by the authors for comparison are models with well-performing public implementations and pre-trained weights. By training the proposed model with the public dataset(eg, LJSpeech), authors can easily compare it with other models. This is the fairest and most reliable comparison method that can claim that the proposed method shows state-of-the-art performance.
>
> The LJSpeech dataset is an excellent public dataset, but does not meet our needs.
>
> 1. Firstly, the prosody diversity of LJSpeech is relatively weak, and it cannot demonstrate the capability of prosody modeling. Our internal data has rich prosody diversity, which puts forward higher requirements for the prosody modeling capability of the model.
> 2. Secondly, the sound quality of LJSpeech is relatively poor, with reverberation and high frequency breaking. To explore the upper bound of the generated audio quality, we choose an internal dataset. Our internal dataset has high-quality recordings and 48k audio, which can fully demonstrate the model's capability of waveform generation.
>
> > Some figures need to be improved, such as font size.
>
> Thank you for your suggestion.
>
> ### Summary
>
> In particular, our paper proposes a new prosody modeling method. We model the phoneme-level prosody in an implicit way, assuming that the prosody obeys a non-standard normal distribution. It cannot fully characterize prosody information to use pitch or energy to model prosody explicitly such as FastSpeech 2, FastPitch and etc. In addition, we use the prosody predictor to predict the prosody distribution instead of the fixed prosody vector, which can avoid the over-smoothness of the prosody prediction, because for the same text, there will be different prosody distributions. Therefore, we assume that prosody obeys a non-standard normal distribution, which increases the upper bound of prosody representational power, and CLONE's prosody predictor avoids the over-smoothness of prosody prediction, improving the expressiveness of the generated audio in end-to-end synthesis.
>
> We hope our replies solve all your concerns about the paper. If we successfully address your concerns, we would strongly appreciate an increased score; otherwise, we are happy to provide additional discussion and address any further questions. Thanks!

---

> ### Author Response · Authors · 2022-08-08
> **Reply to Reviewer 1KcR**
>
> Dear Reviewer 1KcR:
>
> We thank you for the precious review time and valuable comments. We have provided corresponding responses and results, which we believe have covered your concerns. We hope to further discuss with you whether or not your concerns have been addressed. Please let us know if you still have any unclear parts of our work.
>
> Best.

---

### Official Review · Reviewer_Pq1m · 2022-07-14

**Rating:** 8
**Confidence:** 3
**Soundness:** 4 excellent
**Presentation:** 4 excellent
**Contribution:** 4 excellent

**Summary:**

The paper introduces a novel approach for prosody control in text-to-speech models. A sequence of prosody representations are introduced as latent variables in variational training. Different from prior work, where prosody information are explicitly represented, the propose method learns general and implicit ones. The general representation, on the other hand, enables better modeling of prosody information, which is crucial for on-to-many mapping problem in TTS systems. The paper also provides numerical results compared with recent works, showing a significant improvements on naturalness. The authors also provides system outputs for baseline and proposed model, which shows advantages of the proposed model.

**Questions:**

1. Could you add variable notations in Figure 1 (e.g. where are x, c, z etc)
2. What do you mean by "severe linguistic information leakages" in line 118. Could you illustrate more?
3. How do you get the hard alignment matrix A in equation (2)? I found a term called hard alignment prosody extractor in section 4.5.3, could you illustrate more?
4. In equation (4), what is normalizing flow f_theta? Is there a reference to such design of priori? If not, could you illustrate more?
5. There are six weights in loss functions and the values are pretty diverse according to appendix. And only one set of values are provided. Is the model performance sensitive to weight values?
6. What does CMOS stands for on line 276?
7. In section 4.5.2, Prosody Variation, you mentioned a high variance of from standard normal distribution sampling. I was wondering if you have considered the mismatch between training and inference (mode(b) in this case). During the training time it seems that the priori is not standard.
8. This is not question but personal opinion, which is not related to the quality of the paper. While I understand the authors would like to choose a good abbreviation, (CLONE in the paper), the paper title is not super related to the novelty and strength of model, which is prosody modeling. (controllable and lossless sound very generic)

**Limitations:**

The authors address their concern on limitations and potential negative societal impact.
As to the limitations, I would suggest author provide more training details, especially on hyperparameter tuning, which can be super useful for future research to replicate the work.



**Strengths And Weaknesses:**

Strengths
1. The paper in general is well written. The introduction of methodology and motivation is straightforward.
2. The methods, including prosody modeling and dual parallel auto-encoder, are fairly novel.
3. The numerical results and system output support the advantage of proposed model.
4. The analysis and ablation study are thorough.

Weaknesses
I don't see significant weaknesses in the paper. However, there are some minor issue I will later address in the question section.

---

> ### Author Response · Authors · 2022-08-02
> **Reply to Reviewer Pq1m (1)**
>
> Thanks for your comments on our paper. We reply to your questions as follows:
>
> > Could you add variable notations in Figure 1 (e.g. where are x, c, z etc)
>
> $c $ and $ z $ denote the linguistic feature and the phoneme-level prosody latent variable respectively, and $ x $ denotes a data point in our training dataset. $ c $ is the "Linguistic Feature" in the figure and $ x $ is the spectrogram in the figure. $ z $ isn't visualized in the figure but it is the result of "Reparameterization". $ ir $ and $ \hat{ir} $ is from acoustic encoder and posterior wave encoder respectively. $ \mu_{pp}, \sigma_{pp} $ and $ \mu_{\phi}, \sigma_{\phi} $ is the predicted distribution from prosody predictor and posterior encoder respectively. $ PF_{phone} $, $ PF_{frame} $ and $ \widehat{PF}_{frame} $ denote phoneme-level prosody feature, frame level prosody feature and frame-level prosody feature expanded from $ z $.
>
> > What do you mean by "severe linguistic information leakages" in line 118? Could you illustrate more?
>
> Linguistic information leakage here means that frame-level prosody feature $PF_{frame}$ is not downsampled, so it contains too much information causing the acoustic encoder predicts intermediate representations directly from $ PF_{frame} $ without the use of linguistic features (just like the linguistic information leaks through $PF_{frame}$) during the training of the model. Note that the prosody modeling method (as shown in Figure 1 (b)) is based on VAE, so the downsampling operation is critical to extract prosody information and filter out linguistic information. Thus, we extract the prosody feature at the frame level and downsample it to the phoneme level before putting it into the acoustic encoder.
>
> > How do you get the hard alignment matrix A in equation (2)? I found a term called hard alignment prosody extractor in section 4.5.3. Could you illustrate more?
>
> During training, we get the hard alignment matrix $\textbf{A}$ from the ground-truth durations $\mathbf{d} = \{d_1, ..., d_{T_p}\} $ of phonemes. $\mathbf{A} \in \mathbb{R}^{T_p \times T_f}$ and each element in A is 0 or 1.
>
> $A_{ij} = 1$, when $\sum_{k=1}^{i-1}d_k \leq j < \sum_{k=1}^{i}d_k$
>
> $A_{ij} = 0$, when otherwise.
>
> In other words, $A_{ij} = 1$ denotes The i-th phoneme is being pronounced at the j-th spectrogram frame. Hard alignment prosody extractor means we extract prosody by using the hard alignment matrix $\mathbf{A}$ rather than a soft attention mechanism e.g. parallel tacotron[1], which represents the operation of converting $PF_{frame}$ to $PF_{phone}$.
>
> > In equation (4), what is normalizing flow f_theta? Is there a reference to such design of prior? If not, could you illustrate more?
>
> The normalizing flow $f_{\theta}$ (Real NVP[2]) is a stack of affine coupling layers consisting of a stack of non-causal WaveNet residual blocks. Same as VITS[3], we simply the normalizing flow with a volume-preserving transformation with the Jacobian determinant of one.
>
> > There are six weights in loss functions and the values are pretty diverse according to appendix. And only one set of values is provided. Is model performance sensitive to weight values?
>
> The efficiency of prosody modeling is sensitive to the coefficient of reconstruction loss $L_{recon}$, which is $ \lambda_{1} $. Thus, we increase the $ \lambda_{1} $ to ensure the modeling of prosody and we use the same coefficient as HiFi-GAN[4], which is 45.0. The other coefficients of loss functions are not sensitive to model performance.
>
> Reference
>
> - [1] Parallel Tacotron: Non-Autoregressive and Controllable TTS.
> - [2] Density estimation using Real NVP.
> - [3] Conditional Variational Autoencoder with Adversarial Learning for End-to-End Text-to-Speech.
> - [4] HiFi-GAN: Generative Adversarial Networks for Efficient and High Fidelity Speech Synthesis

---

> ### Author Response · Authors · 2022-08-02
> **Reply to Reviewer Pq1m (2)**
>
> > What does CMOS stand for on line 276?
>
> We conduct CMOS evaluation ([speech quality assessment](https://ecs.utdallas.edu/loizou/cimplants/quality_assessment_chapter.pdf)) which compares the voice quality between two voices. The test taker will rate the two audios with a score of -2, -1, 0, 1, and 2, respectively. The higher the score, the higher the quality of the test audio than the reference audio. We let the 48 kHz ground-truth audio be compared with the samples of 48 kHz CLONE, 24kHz CLONE and 24kHz VITS respectively. The CMOS evaluations demonstrate that 48 kHz CLONE can generate the audio close to the 48 kHz ground-truth audio and outperforms both the 24 kHz CLONE and 24 kHz VITS.
>
> > In section 4.5.2, Prosody Variation, you mentioned a high variance of from standard normal distribution sampling. I was wondering if you have considered the mismatch between training and inference (mode(b) in this case). During the training time, it seems that the prior is not standard.
>
> During training, posterior encoder predicts the phoneme-level prosody distributions from the input linear spectrogram and the condition linguistic feature, which aren't standard normal distributions but non-standard normal distributions. Then we conduct a normalizing flow to transform the non-standard normal distribution to the standard normal distribution, thus, with the invertible transformation of flow model, we can sample value from the standard normal distribution and transform it to the non-standard normal distribution. In mode b inference, we sample value from the standard normal distribution and transform it to the non-standard normal distribution by flow model.
>
> > This is not question but personal opinion, which is not related to the quality of the paper. While I understand the authors would like to choose a good abbreviation, (CLONE in the paper), the paper title is not super related to the novelty and strength of model, which is prosody modeling. (controllable and lossless sound very generic)
>
> Thank you very much for your suggestion. We will consider using new model name in next work.

---

### Meta-Review · Area_Chair_TZMA · 2022-09-09

**Recommendation:** Reject
**Confidence:** Certain

**Metareview:**

I am in agreement with the last 2 reviewers.

1) there are many concerns about the technical correctness of the paper that can be improved
2) more thorough evaluations and experiments are needed

i'm marking this as reject and i encourage the authors to address reviewer comments and resubmit.

**Award:**

No

---

### Decision · Program_Chairs · 2022-09-14

Reject